# Data Collection in an IoT Off-Grid Environment Systematic Mapping of Literature

**DOI:** 10.3390/s22145374

**Published:** 2022-07-19

**Authors:** Ademir Goulart, Alex Sandro Roschildt Pinto, Adão Boava, Kalinka Branco

**Affiliations:** 1Computer Science Graduate Program, Federal University of Santa Catarina, Florianópolis 88040-370, Brazil; a.r.pinto@ufsc.br (A.S.R.P.); adao.boava@ufsc.br (A.B.); 2Institute of Mathematics and Computer Science, Universidade de São Paulo, São Carlos 13566-590, Brazil; kalinka@icmc.usp.br

**Keywords:** data collection, IoT Off-Grid, drone, data mule, TSP, OMNeT++

## Abstract

The goal of this work is to present a systematic literature mapping (SLM) identifying algorithms for the search for data, determining the best path and types of communication between the local server and the drone, as well as possible simulators to validate proposed solutions. The concept, here considered as IoT Off-Grid, is characterized by being an environment without commercial electrical infrastructure and without communication connected to the internet. IoT equipment generates data to be stored on a local server. It collects these data through a drone that searches each local server for later integration with the commercial internet environment. As a result, we have algorithms to determine the best path based on the TSP—travelling salesman problem. Different types of communication between the drone and the server contain the data, predominantly WiFi 802.11. As a simulator, OMNeT++ stands out.

## 1. Introduction

The concept considered here as IoT Off-Grid, Figure 1, represents an environment where sensors and controllers are installed in a place totally devoid of electricity and communication. Therefore, it is necessary to generate electricity to supply the site. This generation can be obtained with the use of photovoltaic equipment, wind energy, or another form of own generation of electric energy. A photovoltaic system is neededwith batteries sized for the load needed by the environment and an inverter to meet the energy demand in the IoT Off-Grid environment.

The information generated in this IoT Off-Grid environment will be collected periodically through a data mule (DM), in this case a drone, also identified in the literature as an unmanned aerial vehicle (UAV). The flight autonomy of the drone will be considered, along with selection of the best route, as well as the age of the information collected, considering the time of generation in the environment and the time of collection by the drone. A scenario is shown in Figure 2.

### 1.1. SLM Research Objectives and Questions

The main objective of this study is to systematically map current research on data collection using a data mule in a geographically dispersed IoT. Considering that the data will be collected through a drone in one or more local networks with several sensors and a data concentrator of this local network, these local networks are isolated in environments devoid of internet and with local infrastructure to this environment. For the simulation of the proposed model, a network simulation software suitable for the environment will be used.

In this sense, this SLM also aims to identify, categorize, and synthesize important studies in this area.

As research questions (RQ), there are the following:RQ1—What algorithm is used in routing for data collection?RQ2—What technology is used for the reception of data by the drone?RQ3—What network simulator software was used in the study?

### 1.2. Article Organization

In Section 1 we provide, in addition to the introduction, the objectives and research questions of the SLM. In Section 2, the planning for the execution of the SLM is presented. The search string and search sources with selection procedures are highlighted, as well as inclusion and exclusion criteria. Section 3 describes the SLM execution process with the steps of inclusion, selection, and evaluation. Backward and forward snowballing procedures are also presented. Section 4 presents the results obtained based on the specified requirements. Section 5 describes the works selected in this study and Section 6 presents the conclusions of this SLM.

## 2. Planning

### 2.1. Search String and Research Sources

To successfully search for important studies, search terms are critical. In a study by Keele [1], the author recommended considering the perspectives population, intervention, comparison, and outcome (PICO). These perspectives have been widely used by many systematic mapping studies [2,3]. However, in this study, as far as the general rationale of the PICO framework is concerned, we built a generic search string to support search stability across many databases. Thus, to conduct the search in the data sources, according to Table 1, in a generic way, the search string serves as a guide.

STRING: **IoT AND (UAV OR drones) AND "*****data collection*****"**

### 2.2. Inclusion and Exclusion Criteria

In order to answer the research questions in this SLM, inclusion and exclusion criteria were formulated and used.

The inclusion criteria were the following:IC1—Studies that focus on collecting data generated in IoT with drone collection.IC2—Papers with drone routing for data collection.IC3—Relevant articles in the last years.IC4—Works using network simulator.

The exclusion criteria were the following:EC1—Studies not associated with the research questions.EC2—Duplicate articles where the same topic was being evaluated.

## 3. Execution

### Selection Procedure

The selection procedure aims to identify studies that are significant for the purpose of the SLM. Figure 3 shows the sequence of steps for this selection procedure. The execution of the search in the different bases, from the search string, presented the result according to Table 2.

In the initial identification procedure from the results from all the bases, a removal of duplicate items was performed, removing the same items retrieved in different bases, eliminating 31 items. In the next phase of selection, for a total of 552 items, considering an analysis based on titles and abstract, 497 items were excluded. There remained 55 items to be evaluated, according to their full text and based on the inclusion and exclusion criteria. In the full text evaluation stage, the inclusion and exclusion criteria were considered, and thus a total of 39 were excluded. In the end, 16 works remained to be analyzed. During the analysis of these 16 works, the backward and forward snowballing process was performed for each of the [4] works.

In the snowballing backward analysis, 16 new works were identified and in the forward, 8 new works that could be interesting. After a more detailed reading of the 24 articles from snowballing, 3 works were selected in cycle 1 of snowballing. From these 3 selected works, a second snowballing cycle was carried out. In the backward snowballing analysis, 3 works were identified, and in the forward, 5 works that could be interesting after this second snowballing cycle. Some references were repeated and previously selected. After this second cycle, only 1 reference was selected and added to the already selected ones.

Considering this work selected in cycle 2, a third snowballing cycle was performed. In the snowballing backward analysis, 4 works were identified, and in the forward analysis, no work was identified. After this third cycle, no reference was selected to be added to the others already selected. At the end of the complete process, from the initial selection and the different snowballing cycles, 20 works were found to be considered for data extraction. The list of the 20 selected papers is presented in Table 3.

## 4. Requirements Results

An overview presenting a word cloud, based on the content of all titles of all selected works, is presented in Figure 4. Considering the search string encompasses IoT and data collection and UAV, these terms can be identified in the word cloud in a prominent way in relation to the others. Other words also stand out in the cloud, such as algorithm, networks, planning, and internet, among others with less intensity.

Among the 20 selected works, 9 are from conference papers and 11 are from journal articles. Regarding the date of publication, Figure 5 presents the number of publications in each year in the period from 2017 to 2021 (until August).

The distribution of the authors of the works considering the country of their institutions is presented in Table 4 and in Figure 6.

Considering the institutions, the distribution of authors, according to their institutions, is shown in Figure 7.

After reading and evaluating all the works selected in the previous phase, the results related to the different research questions of this SLM are presented.
RQ1—What algorithm is used in routing for data collection?RQ2—What technology is used for the reception of data by the drone?RQ3—What network simulator software was used in the study?

### 4.1. RQ1—Algorithm Used in Routing

The predominant algorithm in the evaluated works to define the shortest path to be taken by the drone during data collection is the traveling salesman problem (TSP) algorithm, a well-known method to determine the shortest path to reach different points with the least distance. For a total of 11 works that present routing algorithms, 5 deal with TSP. TSP can be implemented with a heuristic method for a finite number of points reached by one drone. Another algorithm that stands out in the studies is the one that deals with the path in an ant colony, known as ant colony algorithm, or TSP-ant colony optimization (ACO). The ANT algorithm is found in two studies. A list of the algorithms used in routing is presented in Table 5. Regarding this RQ1, it is important to note that of the 20 selected works, 11 presented specific routing algorithms while the others presented prototypes with data collection without routing algorithms. Additionally, three survey/review papers on routing algorithms were selected and are included in the total of 20 final papers.

### 4.2. RQ2—Technology Employed to Receive Data

Regarding the technology used for data collection, the works that presented practical cases with prototypes always pointed out which technology was used for data collection. Some works with mathematical models and theoretical simulations did not deal with communication for data collection. A list of the technologies used to receive the data, referring to 11 works, is presented in Table 6.

### 4.3. RQ3—Network Simulator Software

Of the 20 evaluated works, 6 presented simulations of the models with the presentation of the results found after simulation. A list of the different simulators found in the works is presented in Table 7.

## 5. Work Description

In this systematic mapping of the literature, several works related to data collection using drones stand out. In Gagliarde et al. [5], there is an agro-meteorological data collection system using a Bluetooth Low Energy (BLE) transceiver as communication. It presents a prototype with its configuration used for the experiment.

Wang et al. [6] present a precision adjustable trajectory planning (PATP) scheme. The paper also considers the power consumption of wireless communication in the on-demand PATP (OD-PATP) scheme.

For Cao et al. [7], the communication used the ZigBee 2.4 GHz protocol and the algorithm to determine the collection path based on an optimization algorithm such as the ant colony algorithm. He presented an experiment with collection at four centralizing points.

In Yang et al. [8], research on the main problems in IoT data collection using drones is presented. The paper considers the possible ways of grouping sensors into clusters. Regarding the planning of drone paths, they cite several algorithms: initially as the traveling salesman problem in the fast path planning with rules (FPPWR) algorithm, an alternative formulating the movement between sensors as a Markov chain. The use of Q-learning neural network, A-star algorithm, and genetic algorithms are also considered in the work.

A scheme of waking the sensor only when it is going to transmit is presented in Trotta et al. [9], and they propose a framework called BEE-DRONES for large-scale wireless sensor networks. It considers the optimal trajectory together with the lifetime of the sensors. The problem is transformed into a multiple-variable optimization with a centralized or distributed heuristic solution over multiple graphs. It uses OMNeT++ as a simulator [25].

In Goudarzi et al. [10], the BL-TSP algorithm is pointed out to define the search path between the data collection points. Besides the classic traveling sales problem (TSP), it considers Bezier curves so that it smooths the paths.

Zhang and Li’s work [11] does not consider routing, but considers the collection of a remote point with sensor activation via LoRa and data transmission using IEEE 802.11 ac. A use case is presented with equipment based on specific hardware, assembled to meet the project.

In Min et al. [12], a dynamic collector node scheme is proposed, where the determination of the collector node takes into account the drone speed and latency in data collection without predetermined information. A three-tier protocol is presented in Quin et al. [13]. An application layer over a modified implementation of ContikiMAC and over IEEE802.15.4 2.4 GHz. This protocol is called UIWP (UAV integrated WSN protocol) and the experiment considered the effects of speed, altitude, and approach angle of the node with the data to be collected. They considered a drone and a node at work.

In the work of Safia et al. [14], a distributed algorithm considering energy efficiency is proposed, called HCS (Hilbert-order collection strategy). The mobile collector establishes the path based on the Hilbert values of the node locations. Potter et al. [15] present the use case for collecting data in monitoring an environment with temperature, pH, and conductivity sensors. They detail the construction of the collector node and the hardware embedded in the drone for communication and data collection.

In Liang et al. [16], a data collection scheme using a protocol based on DTN (delay tolerant network) is proposed. To determine the path, an algorithm using a Hilbert curve is proposed. As an emulation tool, CORE (Common Open Research Emulator) was used.

The LoRa communication is presented in Behjati et al. [17] as a facilitator for the use of drones in monitoring large-scale livestock on rural farms. For the optimization of the drone path, a genetic algorithm is proposed, with an optimization using ant colony algorithm called EPSO (enhanced particle swarm optimization). It uses the LoRaWAN technology with multichannel as communication. The paper presents details of the implementation of the practical case applied to a rural farm.

A review of different techniques in Aggarwal and Kumar [18] is found to plan drone routing. Planning techniques are classified into three categories: representative techniques, cooperative techniques, and non-cooperative techniques. Representative techniques are (a) probabilistic scripts; (b) random trees of fast exploration; (c) Voronoi diagram; (d) A-star algorithm. Cooperative techniques are presented: mathematical models; bio-inspired models; machine learning models; multi-objective optimization models.

In Lima et al. [19], a work on data collection carried out in the forest is presented. Using 802.11 communication, the study evaluates the performance of this communication in collecting photographic images of animals collected by cameras installed in the forest. The work presented by Abdelhamid [20] considers data collection in an environment with damaged or inoperative infrastructure due to emergency situations. It considers, as determining the path for the drone, a generalization of the traveling salesman problem. It considers three planning schemes: (a) coverage first; (b) priority first; (c) balanced.

An algorithm based on the simple division of the area to be visited is proposed by Medani et al. [21] and the results are measured based on the NS3 simulator [26].

In the work of Xu and Che [22], a brief review of algorithms to solve the traveling salesman problem is presented. In Liu et al. [23], the routing algorithm considers the age of the collected information as a priority. Both a solution based on dynamic programming and a genetic algorithm are presented. Another work that considers the age of information is presented in Changizi and Emadi [24], not for data collection but for updating data at points with IoT. It considers a prioritization in the selection of points, based on the knapsack problem combined with the traveling salesman problem.

The work of Wang et al. [27] explores vehicular crowdsourcing (VC) by a group of unmanned vehicles (UVs) such as drones and unmanned ground vehicles to collect these data from points of interest (PoIs). They explicitly consider to navigate a group of UVs in a three-dimensional (3D) disaster work zone to maximize the amount of collected data, geographical fairness, and energy efficiency while minimizing data dropout due to limited transmission rate.

A study of the mobile access point deployment in workflow-based mobile sensor networks is provided by Jin et al. [28]. They categorize mobile users (MUs) workflows according to a priori knowledge of MUs’ staying durations at mission locations into complete and incomplete information workflows. They formulate the cost-minimizing mobile access point deployment problem in both categories into multiple (mixed) integer optimization problems, satisfying MUs’ QoS constraints.

A novel drone-based collaborative sparse-sensing framework droneSense is proposed by Zhao et al. [29]. droneSense selects a minimum number of points of interest (POIs) to schedule drones for physical data sensing and then infers the parking occupancy of the remaining POIs to meet the overall quality requirement.

## 6. Conclusions

In this SLM, it can be seen that the interest in collecting data with the use of drones has grown in recent years, although the number of publications in 2021 is only partial, up until the month of August 2021. The two countries that stand out the most in this survey are China and the United States. The largest number of publications are from authors linked to universities such as School of Cyber Science and Engineering Southeast University, James Madison University, Universiti Kebangsaan Malaysia, and University of Bologna. A total of eight works present prototypes with practical experiments while a total of ten present mathematical models. With the use of simulators, six works are found. The most popular simulators in these works are OMNeT++ [25] and NS3 [26]. Regarding the communication technologies found in the works, ZigBee, Bluetooth, LoRa, and WiFi 802.11 can be highlighted. In the works evaluated, several algorithms for routing were addressed. The traveling salesman algorithm or TSP (traveling salesman problem) predominates, with solutions in different ways and variations of restrictions such as the drone autonomy limit and the age of the collected data. All works that were presented refer to data collection in wireless sensor networks or in isolated sensors. The IoT Off-Grid concept considers a site isolated from the commercial energy and communication infrastructure. This site considers the generation of local electrical energy, sensors and actuators connected in a local network, a server for low-cost network management and data collection, and access point for the wireless connection with the drone.

The contributions of this MSL can be summarized as mapping the literature for:Algorithm to determine the best path for the drone to go to the points.Type of communication between the drone and the server containing the data.Network simulator.

## Figures and Tables

**Figure 1 sensors-22-05374-f001:**
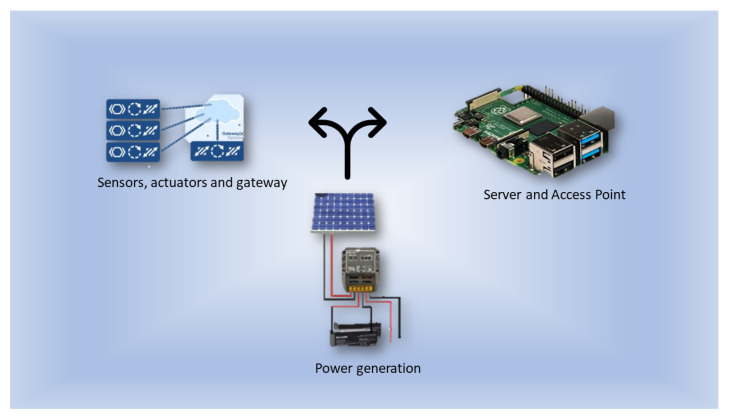
Station Off-Grid.

**Figure 2 sensors-22-05374-f002:**
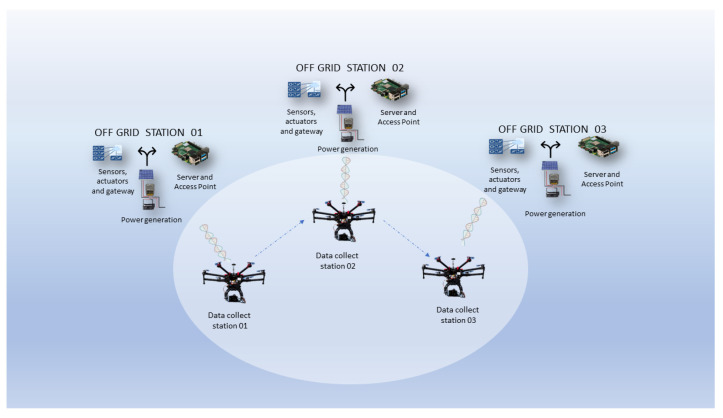
Scenario.

**Figure 3 sensors-22-05374-f003:**
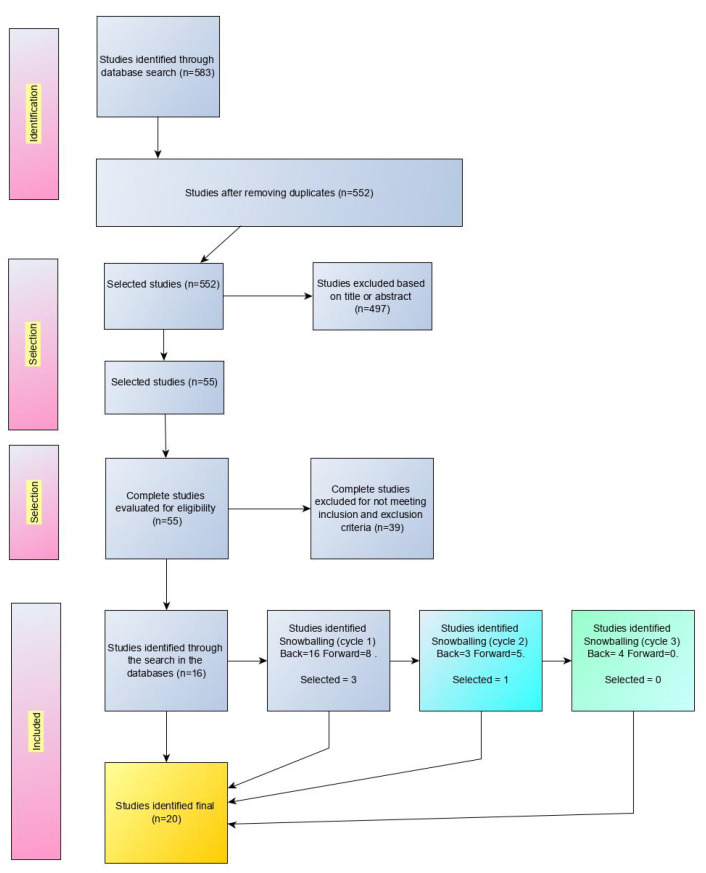
Selection procedures.

**Figure 4 sensors-22-05374-f004:**
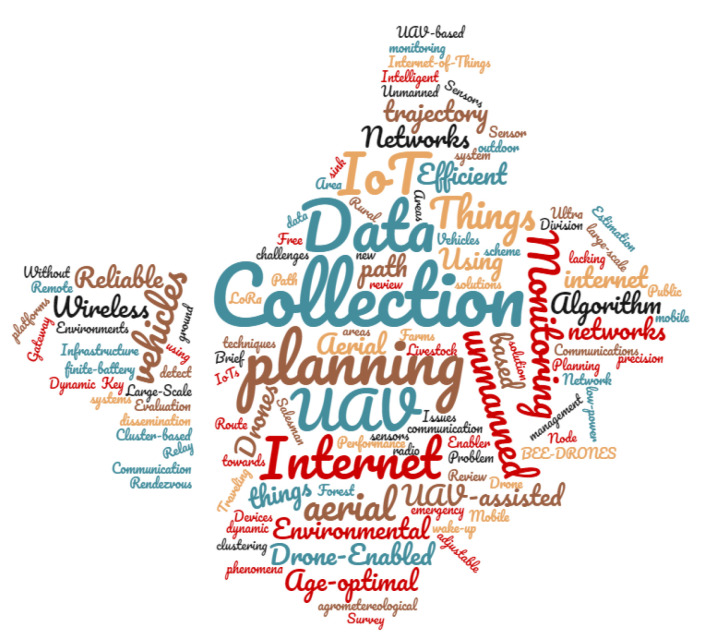
Word cloud—paper title.

**Figure 5 sensors-22-05374-f005:**
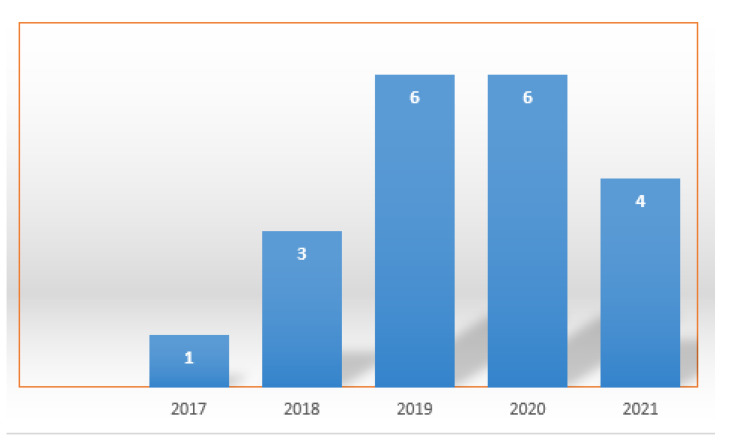
Date of publication.

**Figure 6 sensors-22-05374-f006:**
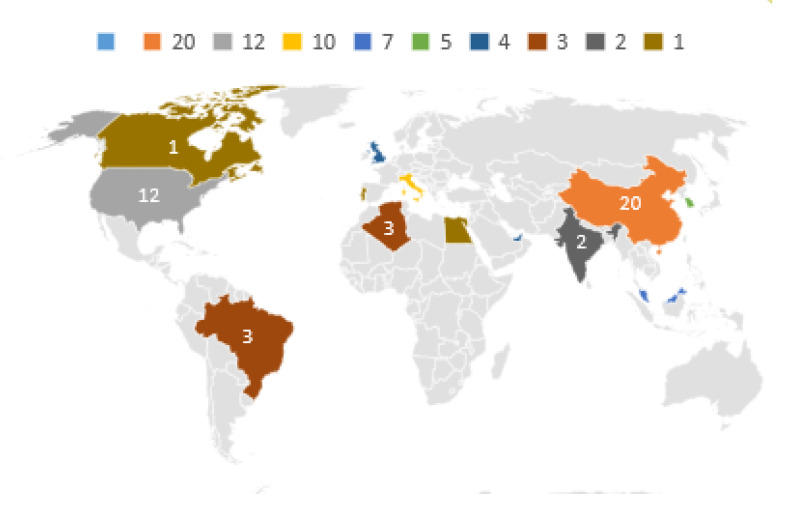
Country distribution.

**Figure 7 sensors-22-05374-f007:**
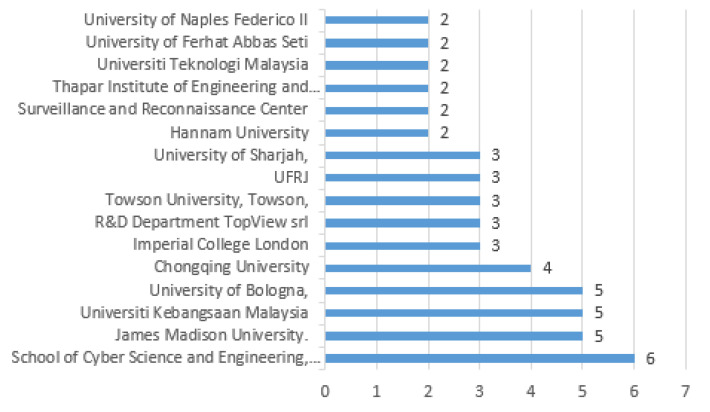
Distribution by institutions of the authors.

**Table 1 sensors-22-05374-t001:** Data sources.

Name	Link
**IEEE Xplore**	http://ieeexplore.ieee.org/
**Science Direct**	http://sciencedirect.com/
**ACM**	http://dl.acm.org/
**Springer Link**	http://link.springer.com/
**Wiley**	http://onlinelibrary.wiley.com/
**MDPI**	https://www.mdpi.com/
**SCOPUS**	https://www.scopus.com/

**Table 2 sensors-22-05374-t002:** Data source recovery.

Data Source	Recovered Items
**IEEE Xplore**	16
**Science Direct**	170
**ACM**	3
**Springer Link**	237
**Wiley**	5
**MDPI**	7
**SCOPUS**	145
**TOTAL**	583

**Table 3 sensors-22-05374-t003:** Selected papers.

Reference	Title
[5]	A new system for agrometereological data collection in areas lacking communication networks
[6]	A precision adjustable trajectory planning scheme for UAV-based data collection in IoTs
[7]	A solution for data collection of large-scale outdoor internet of things based on UAV and dynamic clustering
[8]	A Survey of Key Issues in UAV Data Collection in the Internet of Things
[9]	BEE-DRONES: Ultra low-power monitoring systems based on unmanned aerial vehicles and wake-up radio ground sensors
[10]	Data collection using unmanned aerial vehicles for Internet of Things platforms
[11]	drone-Enabled Internet-of-Things Relay for Environmental Monitoring in Remote Areas Without Public Networks
[12]	Dynamic Rendezvous Node Estimation for Reliable Data Collection of a drone as a Mobile IoT Gateway
[13]	Efficient and Reliable Aerial Communication with Wireless Sensors
[14]	Efficient data collection by mobile sink to detect phenomena in internet of things
[15]	Environmental Monitoring Using a drone-Enabled Wireless Sensor Network
[16]	Internet of Things Data Collection Using Unmanned Aerial Vehicles in Infrastructure Free Environments
[17]	LoRa Communications as an Enabler for Internet of drones towards Large-Scale Livestock Monitoring in Rural Farms
[18]	Path planning techniques for unmanned aerial vehicles: A review, solutions, and challenges
[19]	Performance Evaluation of 802.11 IoT Devices for Data Collection in the Forest with drones
[20]	UAV path planning for emergency management in IoT
[21]	Area Division Cluster-based Algorithm for Data Collection over UAV Networks
[22]	A Brief Review of the Intelligent Algorithm for Traveling Salesman Problem in UAV Route Planning
[23]	Age-optimal trajectory planning for UAV-assisted data collection
[24]	Age-optimal path planning for finite-battery UAV-assisted data dissemination in IoT networks

**Table 4 sensors-22-05374-t004:** Country distribution.

Country	Quantity
China	20
USA	12
italy	10
Malaysia	7
South Korea	5
U.K.	4
UAE	4
Algeria	3
Brazil	3
India	2
Canada	1
Egypt	1
Hong Kong	1
Iraq	1
Portugal	1

**Table 5 sensors-22-05374-t005:** Algorithm used in routing.

Reference	Algorithm
[6]	*PATP-Precision adjustable trajectory planning*
[7]	*Ant colony algorithm*
[9]	*TSP-ant Colony Optimization (ACO)*
[10]	*BL-TSP algorithm*
[14]	*Path based on the order of Hilbert values*
[16]	*Hilbert-Curve-based path planning algorithm*
[17]	*(TSP) and enhanced particle swarm optimization (EPSO)*
[20]	*Generalization of TSP*
[21]	*Simple area division cluster-based algorithm (SAD-CA)*
[23]	*Max-AoI-optimal and Ave-AoI-optimal*
[24]	*Stage-WSHP*

**Table 6 sensors-22-05374-t006:** Types of communication.

Reference	Communication Technology
[5]	*Bluetooth Low Energy (BLE)*
[7]	A *ZigBee wireless 2.4 GHz*
[9]	*Simple request/replay subGHz radio*
[10]	802.11b (no simulador)
[11]	*LoRa e IEEE 802.11 ac (5ghz)*
[12]	Simulador com IEEE 802.15.4
[13]	*ContikiMAC over the IEEE 802.15.4 2.4 GHz*
[15]	*WiFi*
[16]	*Device with the DTN protocol implemented*
[17]	*Multi-channel LoRaWAN^®^ gateway*
[19]	*WiFi 802.11 2.4 GHz*

**Table 7 sensors-22-05374-t007:** Types of simulator.

Reference	Simulator
[9]	*OMNeT++*
[10]	*OMNeT++; MATLAB e MiXim*
[12]	*OMNeT++*
[14]	*NS2*
[16]	*CORE, v5.1*
[21]	*NS3*

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
