# Peer review of "Data Collection in an IoT Off-Grid Environment Systematic Mapping of Literature"

_sensors, 2022, doi:10.3390/s22145374_

Round 1
Reviewer 1 Report
The author identifies, categorizes and synthesizes important studies on data collection using Data Mule and drone in geographically dispersed IoT.
However, there are several weaknesses in the paper, which make the paper can not be fully appreciated. The shortcomings are identified as follows.
(1) The opportunities and the facing challenges on data collection using Data Mule and drone in geographically dispersed IoT should be supplemented in the paper.
(2) A structural comparison between all the mentioned studies should be supplemented in the paper, especially the methods and algorithms used in the mentioned studies.
(3) Please illuminate more clearly about the traveling salesman algorithm and explain why the algorithm predominates in all the studies mentioned in the paper and the advantages compared to other algorithms.
(4) The authors fail to cite several past literatures (i.e., [1-3]) highly related to this work, and clearly discuss the differences between them and this paper.
[1] Energy-Efficient 3D Vehicular Crowdsourcing for Disaster Response by Distributed Deep Reinforcement Learning. KDD 2021
[2] Cost-minimizing Mobile Access Point Deployment in Workflow-based Mobile Sensor Networks, ICNP 2014.
[3] DroneSense: Leveraging Drones for Sustainable Urban-scale Sensing of Open Parking Spaces. INFOCOM 2022
(5) In the inclusion and exclusion criteria part, the exclusion criteria 2 should be changed. Not only the most recently work but also the most important and meaningful work should be selected.
Reviewer 2 Report
In this paper, a data collection in an IoT Off Grid environment systematic mapping of literature is presented.
The paper represents an interesting approach to performing a review, anyway, some improvements are required:
1. The letters in Fig. 1 and 2 are too small and difficult to read.
2. Section 3: More information related to the SLM should be included.
3. In Tab.4 the names of the countries should be written in English.
4. Ref [6] is missing in Table 6.
5. Please check the order of the references. For example [25] is before [22].
Reviewer 3 Report
Dear Authors,
Thanks for your efforts to provide a Systematic Mapping of Literature related to the IoT Off-Grid environment.
After reviewing all the sections of your article, I have decided to recommend a major revision since the current version lacks systematic contribution and the writing must be improved significantly. Please consider the following comments during the revision stage.
1) In Figure 1, why did you define the Server and access point as an electronic board? These two machines are totally different and must be defined as an identifiable icon like wifi/network cell and servers separately. Figure 2 consists of three replicated version of figure 1 that is technically incorrect.
2) You should summarize the contributions of your studying in three bullets so that the reader can understand the main objectives of the study.
3) I believe that you should write a comprehensive review article and add a systematic mapping of the literature as a section of the survey article. The current version does not contain details of algorithms or technologies and the clear architecture of the IoT Off-Grid environment. However, I doubt that even the classification of the Off-Grid environment is not technically right. Because IoT-based systems consist of several network devices and sensors that have many common features in different environments. Please add a section and clarify why did you change the category of the IoT Off-Grid environment from the Smart environment category?
4) You should review +100 existing articles as your reference and summarize their deployed technologies and algorithms. Then, you may add the mapping methodology as a section of a comprehensive review. The current version does not contain enough contributions and can not be accepted as a journal article.
5) On page 3, section 2, you repeated many times the subject "Studies " in the IC1, IC2, and IC4 bullet points as well as these sentences suffer from writing and meaning problems. None of these sentences are written in a formal/official manner. There are plenty of other writing errors that must be proofread by an expert or native scientist in the field.
Round 2
Reviewer 1 Report
The authors have addressed my comments to the previous version. I do not have further comments, and recommend that this paper be accepted.
Author Response
Thank you very much for the review
Reviewer 2 Report
All the requested changes have been performed and the paper has been improved.
Author Response
Thank you very much for the review
Reviewer 3 Report
Dear Authors,
I have checked the revised version of your article and unfortunately did not see considerable changes. I would strongly suggest addressing my former comments. You did nothing to consider my comments. So, I do not think the current version is qualified to be published.
Author Response
Unfortunately, seven days is not enough to make the corrections. I would like to request a new deadline of one month.